# Recent Progress in Flexible Piezoelectric Tactile Sensors: Materials, Structures, Fabrication, and Application

**DOI:** 10.3390/s25030964

**Published:** 2025-02-05

**Authors:** Jingyao Tang, Yiheng Li, Yirong Yu, Qing Hu, Wenya Du, Dabin Lin

**Affiliations:** 1School of Optoelectronic Engineering, Xi’an Technological University, Xi’an 710032, China; 15930440509@163.com (J.T.); liyi_heng2001@163.com (Y.L.); 2School of Ocean Engineering and Technology, Sun Yat-Sen University, Zhuhai 519000, China; huqing3@mail.sysu.edu.cn; 3Media Lab, Massachusetts Institute of Technology, Cambridge, MA 02139, USA; wenyadu@media.mit.edu

**Keywords:** piezoelectric, tactile sensors, flexible, wearable electronics, PVDF

## Abstract

Flexible tactile sensors are widely used in aerospace, medical and health monitoring, electronic skin, human–computer interaction, and other fields due to their unique advantages, thus becoming a research hotspot. The goal is to develop a flexible tactile sensor characterized by outstanding sensitivity, extensive detection range and linearity, elevated spatial resolution, and commendable adaptability. Among several strategies like capacitive, piezoresistive, and triboelectric tactile sensors, etc., we focus on piezoelectric tactile sensors because of their self-powered nature, high sensitivity, and quick response time. These sensors can respond to a wide range of dynamic mechanical stimuli and turn them into measurable electrical signals. This makes it possible to accurately detect objects, including their shapes and textures, and for them to sense touch in real time. This work encapsulates current advancements in flexible piezoelectric tactile sensors, focusing on enhanced material properties, optimized structural design, improved fabrication techniques, and broadened application domains. We outline the challenges facing piezoelectric tactile sensors to provide inspiration and guidance for their future development.

## 1. Introduction

The sense of touch is a crucial modality for human engagement with the external environment, facilitating the perception of physical attributes such as temperature, pressure, and texture [1]. The human skin, being the primary organ of touch, contains many receptor types that transform mechanical stimuli into neural impulses, which are subsequently relayed to the brain for interpretation [2,3]. This perceptual mechanism is crucial for humans and also serves as a source of inspiration for sensor technology development. Tactile sensors represent a significant domain of sensor technology that facilitates the perception of intricate environments, emulating human tactile capabilities and finding extensive use across many sectors.

As application scenarios proliferate, the inflexibility of conventional sensors renders them inadequate, prompting researchers to increasingly focus on flexible sensors. The swift advancement of materials science, flexible electronics, and nanotechnology has significantly enhanced the fundamental attributes of devices, such as sensitivity, range, size, and spatial resolution. Consequently, device design has become more refined, and integration strategies have matured, allowing flexible sensors to conform to various shapes and environments, thereby offering greater design flexibility and application possibilities. Flexible tactile sensors can be categorized based on their operational principles into piezoresistive [4,5,6,7], capacitive [8,9,10,11], piezoelectric [12,13,14,15,16], and triboelectric [17,18,19,20,21]. The most common types are piezoresistive and piezoelectric. The piezoresistive principle is that when an external force is applied to the sensor, the elastomer is deformed and the resistivity changes with the magnitude of the pressure, resulting in a change in resistance that converts the mechanical stimulus into an electrical signal [22,23]. Piezoelectric tactile sensors are sensors based on the piezoelectric effect, which are capable of converting mechanical stimuli into electrical signals, thus realizing the perception and detection of external tactile information, both of which are widely used in the fields of smart homes, industrial manufacturing, health monitoring, and so on [24]. Piezoelectric tactile sensors play a more important role with their unique performance; their primary performance attributes encompass self-generation, significant flexibility, and elevated sensitivity. Their capacity to transform mechanical energy directly into an electrical signal without requiring an external power source renders them advantageous for energy harvesting and low-power applications. The elevated sensitivity and rapid reaction characteristics of piezoelectric materials render piezoelectric tactile sensors particularly appropriate for dynamic measurements [25,26].

In terms of applications, piezoelectric tactile sensors are used in a wide range of fields such as motion monitoring [27,28,29], medicine [30,31], health monitoring [32,33], robotics [34,35], marine technology [36,37], aerospace [38], and e-skin [39,40]. Their accuracy has been markedly enhanced, particularly when integrated with artificial intelligence technologies. In the medical domain, piezoelectric sensors can detect subtle physiological signs, including pulse and respiration rates [16]. In robotics, they can be used as electronic skins to provide feedback about contact and pressure, enhancing the robot’s ability to interact with its environment [41]. For ocean exploration, flexible tactile sensors are used in conjunction with autonomous underwater vehicles to build tactile sensing systems for underwater robots [36].

This work discusses recent findings on flexible piezoelectric tactile sensors (see Figure 1). Section 2 delineates the piezoelectric effect and categorizes piezoelectric materials into inorganic, organic, and composite types. Section 3 and Section 4 delineate the structural design and preparatory integration of flexible piezoelectric tactile sensors. Section 5 examines its primary application scenarios, while Section 6 encapsulates the current research status of piezoelectric tactile sensors and anticipates future developmental prospects for further research.

**Figure 1 sensors-25-00964-f001:**
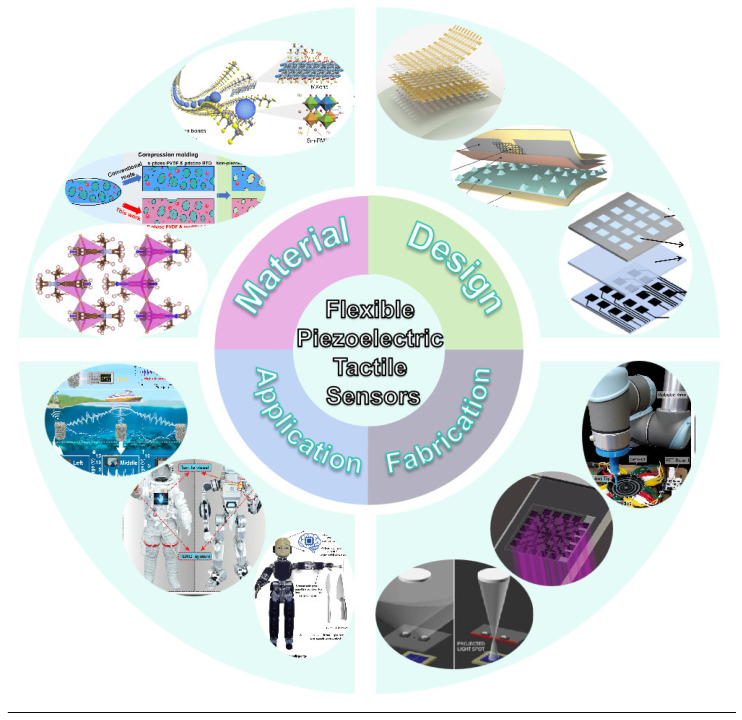
Overview of flexible piezoelectric tactile sensors [42,43,44,45,46,47,48,49,50,51,52,53].

## 2. Piezoelectric Materials

The operational basis of piezoelectric tactile sensors primarily relies on the positive piezoelectric effect exhibited by piezoelectric materials. When the piezoelectric material is subjected to deformation by an external force in a certain direction, its active layer becomes polarized, resulting in the generation of equal quantities of positive and negative charges on the top and lower surfaces. Upon the cessation of the external force, the piezoelectric material will revert to its uncharged state [54]. In order for the piezoelectric material in the sensor to deform and provide an electrical signal, mechanical energy must typically be converted into electrical energy by contact between the object and the sensor.

The piezoelectric coefficient *d*_ij_ is a very important parameter in the physical properties of piezoelectric materials, and the transverse piezoelectric coefficient d_31_ and the longitudinal piezoelectric coefficient *d*_33_ are commonly used to describe the piezoelectric properties of materials [55]. This illustrates the behavior of the connection between the material’s electrical characteristics and mechanical stress, which directly affects how well piezoelectric materials work in the sensor and transducer industries.

### 2.1. Inorganic Piezoelectric Materials

Inorganic piezoelectric materials are inorganic substances with a piezoelectric effect, such as SiO_2_, KNN, PZT, ZnO, MnO_2_, etc., which can convert between mechanical and electrical energy and are widely used in the field of tactile sensors due to their electromechanical coupling effect. Here, we focus on low-dimensional single-crystal materials and piezoelectric ceramics, which are commonly used as pressure-sensitive materials.

#### 2.1.1. Low-Dimensional Materials

Low-dimensional (LD) single-crystal materials in piezoelectric materials mostly refer to materials that have piezoelectric properties. These materials have received a lot of attention because they can be used to make small and flexible electronics that work well. Zheng’s team have made important progress in researching the fabrication of low-dimensional single-crystal materials, which are high-quality single-crystal 2D materials that are the basis for the realization of high-end applications for their electronic and optoelectronic devices [56]. Common low-dimensional single-crystal materials are fullerenes (C_60_), carbon nanotubes, hexagonal boron nitride (h-BN), and MoS_2_. Moreover, ZnO [14,57,58,59,60,61,62], AlN [63,64], and MoS_2_ [65,66,67] are common choices among low-dimensional single-crystal materials due to their excellent material properties, mature preparation processes, and wide application prospects.

Although ZnO’s piezoelectric coefficient (*d*_33_) is low, its outstanding semiconductor and photovoltaic qualities make it a popular material for pressure-sensitive applications, and numerous researchers have been working to improve its performance. In order to improve its electrical output and flexibility, Deng et al. [61] proposed the formation of p-n homojunctions by doping La into ZnO nanorods (NRs) and discrete structural designs, as shown in Figure 2a. They explored and demonstrated the superior performance of the structure in several ways, and the developed discrete ZnO p-n homojunction devices improved the maximum piezoelectric output current and voltage by about 2.3 times compared to the n-ZnO/seed crystal layer devices. Jiang et al. [62] reported a method for the synthesis of aluminum-doped ZnO nanosheets through the controlled synthesis of Al-ZnO nanosheets on a polydimethylsiloxane (PDMS) substrate using multi-walled carbon nanotubes (MWCNTs) as electrodes. The pure ZnO nanorods and Al-ZnO nanosheets were characterized. Shock and bending experiments on pure ZnO nanorods and Al-doped ZnO nanosheets revealed a 216% increase in output voltage for Al-doped ZnO nanosheets. This was due to the excellent properties of aluminum nitride (AlN), such as high resistivity, good coupling coefficient, and low dielectric loss. Zhu et al. [64] chose an AlN thin film as the piezoelectric material for the sensing element and a silicon MOSFET for the microstrain sensor. Applying stress to the AlN-sensitive element generated a charge, which then fed into the MOSFET’s gate, altering the drain–source current and significantly enhancing the sensitivity.

**Figure 2 sensors-25-00964-f002:**
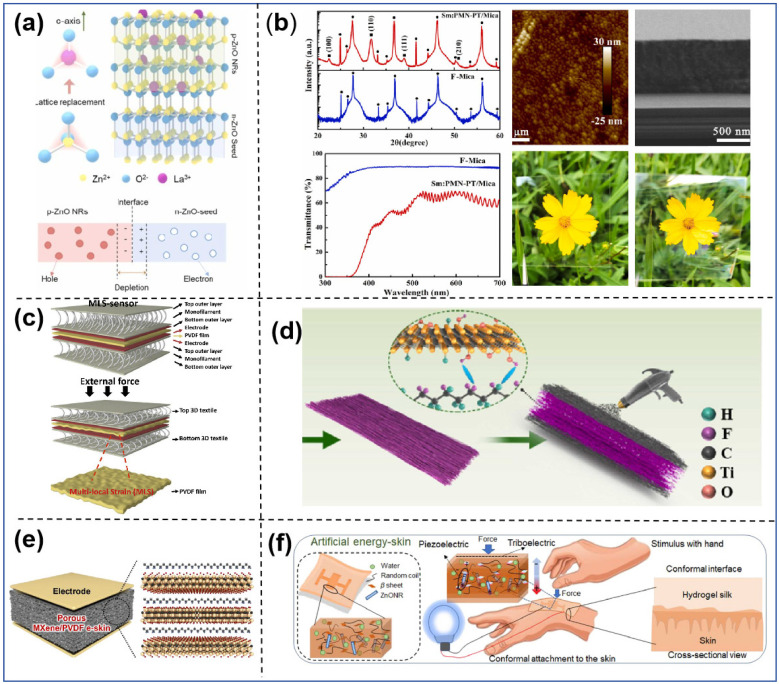
(**a**) Schematic crystal structure of La-doped ZnONR with p-n homojunction [61]. (**b**) XRD patterns, AFM images, and cross-sectional SEM images of the Sm:PMN−PT film and the film appearance [68]. (**c**) MLS sensor structure and (bottom) deformation of the 3D fabric when an external force is applied and there is multi-localized strain in the PVDF film [69]. (**d**) Schematic of PVDF−TrFE nanofibers [70]. (**e**) Schematic illustration of porous MXene/PVDF e-skin and molecular structures of MXene and PVDF [71]. (**f**) Schematic of the concept and working principle of artificial energy-generating skin (EG skin) using silk hydrogel [72].

#### 2.1.2. Polycrystalline Materials

Piezoelectric ceramics generally refer to specially treated piezoelectric ceramic materials, which are usually made from a variety of oxides such as lead oxide (PbO), zirconium oxide (ZrO_2_), titanium dioxide (TiO_2_), etc. and are sintered at high temperatures of 1000 °C to 1300 °C to become polycrystalline [73]. Piezoelectric ceramics have high piezoelectric constants and electromechanical coupling coefficients, making them suitable for high-power and high-sensitivity applications. However, their poor temperature stability, low mechanical strength, poor flexibility, and other shortcomings are the main reasons for their limited application. Barium titanate (BaTiO_3_) [74,75], lead titanate (PbTiO_3_), lead zirconate titanate (PZT) [76], lead niobium magnesium titanate–lead titanate (PMN-PT) [77], and other types are often used in tactile sensors. Niu et al. [78] incorporated PZT particles into solid silicone rubber using a blending process rather than the traditional mixing method, which markedly enhanced the homogeneity of filler dispersion inside the matrix. The proportion of PZT in the composites could be raised to 92 wt% while maintaining an acceptable tensile property of 30%. Lv et al. [68] successfully synthesized Sm-doped PMN-PT (Sm:PMN-PT) transparent films on mica substrates via the sol–gel method, as shown in Figure 2b. The films are very flexible and have a piezoelectric coefficient *d*_33_ of 380 pm V^−1^. This implies their potential application in tracking human movements and creating haptic sensor arrays for touch screens. Meng et al. [13] developed organic thermo-tactile dual-mode transistor sensors using lead zirconate titanate (PZT) piezoelectric ceramics as a substrate, resulting in a simple structure and fabrication process. The prepared devices have high sensitivity, show a linear response to temperature variations from 20 to 60 °C, and are also sensitive to light tapping on the device surface. These devices are a step forward in the practical application of human–computer interaction devices, such as electronic skins and intelligent tactile sensing systems.

### 2.2. Organic Piezoelectric Materials—PVDF and Its Copolymers

In contrast to inorganic piezoelectric materials, organic piezoelectric materials are primarily composed of carbon-based molecules. Sensors, vibrators, and energy harvesting extensively utilize them due to their distinct advantages, which include excellent flexibility, biocompatibility, and cost-effectiveness. Notably, wearable devices employing these materials as pressure-sensitive components have garnered significant interest in the realm of smart medical monitoring, prompting numerous researchers to enhance organic piezoelectric materials. Researchers are actively working to improve the piezoelectric characteristics and stability of organic piezoelectric materials.

It is widely believed that PVDF and its copolymers are the best organic piezoelectric materials because they have excellent mechanical properties, strong chemical stability, great electrical insulation, and a wide range of uses.

#### 2.2.1. PVDF

PVDF is a semi-crystalline ferroelectric polymer with the molecular formula -(CH_2_-CF_2_)_n_-. Five crystalline configurations, α, β, γ, ε, and δ phases, have been found, and only the first three are common. β-phase is the crystalline phase with the highest polarization value and shows the best piezoresponsivity; therefore, the piezoelectric properties of PVDF films are closely related to the content of the β-crystalline phase. Processing conditions such as stretching, polarization, casting, and other treatments, which can transform these crystalline phases into each other, determine the generation of these crystalline configurations [79].

To enhance the performance of tactile sensors made of PVDF materials, it is now necessary to increase the piezoelectric sensitivity and piezoelectric conversion efficiency of PVDF films. Presently, the primary concepts are as follows. The first is to increase the β-phase’s content and enhance its piezoelectric performance through treatments like stretch polarization or by converting the stable α-phase to β-phase using methods like electrostatic spinning in a high-pressure environment [80]. The second concept involves enhancing the internal structure to improve performance, which can be achieved through techniques such as electrostatic spinning, 3D printing, or doping materials with good piezoelectric properties, such as inorganic piezoelectric materials, carbon materials, and other high-performance materials, as detailed in Section 2.3.

Researchers have investigated various aspects of process, structure, and materials with the aim of improving the piezoelectric properties of PVDF to enhance the sensitivity of tactile sensors. Huang et al. [81] proposed a tactile sensor with a rigid–flexible structure in the contact layer to improve the efficiency of force transfer, which triggers the *d*_31_ mode of operation of the PVDF film to work in synergy with the *d*_33_ mode of operation, and the sensitivity of this sensor is 35.6 mV/N, within the linear force detection range of 1–11 N, which greatly improves the sensitivity of the sensor and also provides a new way of thinking. Wang et al. [82] created a tactile sensor by combining a group of PVDF nanorods with an OFET with a bottom gate and a top electrode. The OFET acted as a signal amplifier, which improved the signal-to-noise ratio and also amplified the signal generated by the PVDF, so that the sensitivity of the sensor reached 5.17 kPa^−1^. Ahn et al. [69] proposed a new wearable multi-local strain sensor (MLS sensor) based on a 3D fabric structure, as shown in Figure 2c. MLS sensors utilize pre-strained monofilaments with a three-dimensional fabric structure as pressure sensors to generate amplified strain induced by multi-localized strain (MLS) in PVDF films in order to improve the piezoelectric properties of PVDF. The MLS sensor is shown to amplify the piezoelectric output voltage up to a factor of 5 at the same pressure.

#### 2.2.2. PVDF Copolymers

PVDF has become a functional layer in the majority of tactile sensors due to its flexibility and stability. In order to further improve the piezoelectric properties of PVDF, copolymers of PVDF have been studied, such as polyvinylidene fluoride-hexafluoropropylene (PVDF-HFP) [83,84], polyvinylidene fluoride-trifluoroethylene (PVDF-TrFE) [85,86], and others in order to make it even better at piezoelectricity. Among the many copolymers, the most widely used material for haptic sensors is PVDF-TrFE. Compared with other copolymers, PVDF-TrFE has the advantages of easy polarization and high flexibility. TrFE has the advantages of easy polarization, a high piezoelectric coupling coefficient, good flexibility, and low cost.

An electric field causes the fluorine and hydrogen atoms in the molecular chain to shift relative to one another during the copolymerization of vinylidene chloride (VDF) and trifluoroethylene (TrFE), producing a piezoelectric effect. The VDF content in CVDF is a key factor that affects the performance of PVDF-TrFE. For this reason, learning more about how CVDF affects piezoelectricity is necessary to make the best PVDF-TrFE sensors and energy harvesters. Jiang et al. [87] carried out a full study on the VDF content-guided piezoelectric properties of three commercially available PVDF-TrFE materials. They discovered that the 50:50 film is better for tactile sensing because it has the highest longitudinal piezoelectricity, while the 70:30 film is better for monitoring finger bending because it has the highest transverse electroactivity for monitoring finger flexion. For energy harvesting applications, the 70:30 collector has a maximum power density of 4.96 nW/mm^2^, so the ratio of 70:30 is commonly used in the laboratory.

Based on a 70:30 ratio, combined with production techniques and structural innovations, Yuan et al. [88] used 3D printing to prepare flexible PVDF-TrFE polymer piezoelectric films (PFs) with multiple alternately tilted polarization regions and large polarization lengths in their cross-sectional area. This method differed significantly from conventional methods and greatly enhanced the sensitivity and electromechanical coupling effects of PVDF-TrFE films to detect finger press and finger and wrist joint flexion movement signals. To bring the sense of touch closer to human skin, the researchers combined multiple sensors, while the information fusion was also more complex, requiring the use of high-level algorithms for data processing. Tang et al. [70] have developed a flexible thin-film sensor capable of sensing multidimensional stimuli, as shown in Figure 2d. Spraying MXene on a PVDF-TrFE fiber mat makes the sensor very sensitive. It can also tell when something is stretching in different directions because it has piezoresistive and piezoelectric properties. See Table 1.

**Table 1 sensors-25-00964-t001:** Comparison of the performance of tactile sensors prepared from different piezoelectric materials.

Material	Method	Substrate	Electrode	Sensor Size	Sensor Performance	Reference
ZnO p-n	Spin coating	PEN	Ag	/	Current sensitivity: 82.6 nA/MPa	[61]
BaTiO_3_	Hydrothermal	PDMS	Ti/Au	Area: 3 * 2 cm^2^	Open-circuit voltage: 14 V short-circuit current density: 190 nA cm^−2^	[89]
Al-ZnO	Spin coating	PDMS	MWCNTs	/	Linear: 0.995 Output voltage is increased by 216 percent	[62]
ZnO	Hydrothermal	PDMS	Cu	/	Sensitivity: 1.42 V/N	[60]
AlN	Sputtering	Silicon	Cr-Au	Area: 13,700 μm^2^	GF: 1340	[64]
PZT	Mixing technology	PC	Ag	Area: 5 cm * 4 cm^2^	Power density: 81.25 W/cm^3^	[78]
Sm: PMN-PT	Sol–gel process	PDMS	/	/	Force sensitivity: 5.86 V/N Current density: 150 μA/cm^2^	[68]
PVDF	Pour curing	PDMS	/	Thickness: 28 μm Area: 16 * 16 mm^2^	Sensitivity: 35.6 mV/N Detection range: 1–11 N	[81]
PVDF	Spin coating	PET	Cu	/	Sensitivity: 35.6 mV/N Detection limit: 175 Pa Response time: 150 ms	[82]
PVDF	/	PET	/	/	Sensitivity: 5.17 mV/kPa Responsive time: 150 ms	[69]
PVDF	Spin coating	PDMS	Al	Area: 2 * 2 μm^2^	Sensitivity: 12 mV/kPa Responsive time: 2 ms	[90]
PVDF	3D printing	Conductive sponge	Conductive cloth	Area: 3 * 3 cm^2^	Detection range is 0.1–15 N Responsive time: 80 ms	[91]
PVDF-TrFE	3D printing	PET	Ag	Area: 32 * 10 mm^2^	Sensitivity: 1.47 V/kPa peak power density: 478 μW/cm^2^	[88]
PVDF-TrFE	Spin coating	PI	Al	/	Power density: 4.96 nW/mm^2^	[87]
PVDF-TrFE	Electrostatic spinning	PDMS	Ag	/	Sensitivity: 51.5 mV/N Responsive time: 78 ms (*x*-axis) 46 ms(*y*-axis)	[70]
MoS2-PVDF	Milling and coating	PDMS	Cu	Area: 2 * 2 cm^2^	Power density: 3.2 mW/m^2^ Output: 47 Vpp and 0.12 μA,	[92]
PVDF/ZnO	Electrostatic spinning	PU	Ag	/	Sensitivity: 3.12 mV/kPa Tr/Tf times: 55/75 ms	[16]
PVDF/ZnO	Spinning	PU	Cu	/	Sensitivity: 4.4 mVdeg^−1^, 0.33 V/kPa Response time: 76 ms, 16 ms	[93]
PAN-PVDF hydrogel	/	PDMS	Cu	30 * 8 * 1 mm^3^ and 20 * 20 * 2 mm^3^	Response time: 31 ms Output: 30 mV, 2.8 μA	[94]
Composite hydrogels	One-pot thermoforming, solution exchange	Hydrogels	Conductive tape	/	GF: 19.3 Response time: 63.2 ms	[95]
Silk protein hydrogel	/	PET	Ag	Area: 5 * 5 mm^2^	Power density: 1 mW/cm^2^	[72]

### 2.3. Composite Materials

#### 2.3.1. Composites of Organic and Inorganic Materials

Piezoelectric materials for flexible tactile sensors need to have high piezoelectric constants, excellent stability, and flexibility. Combining inorganic piezoelectric materials with organic piezoelectric materials is the most effective method, and it has become an important research direction for piezoelectric materials. Usually, inorganic piezoelectric materials [96], conductive materials, carbon materials, and other high-performance materials are doped into PVDF and its copolymers to satisfy both the excellent piezoelectric properties and the stability and flexibility of wearable devices.

Kim et al. [71] showed a very sensitive piezoelectric e-skin that could sense a wide range of pressures. They did this by using a porous PVDF structure and adding hierarchical MXene nanosheets as nucleating agents, which made the ferroelectric and piezoelectric properties of PVDF better. The MXene/PVDF e-skin’s porous structure makes it easier for it to deform and concentrate stress in specific areas inside the pores. Yang et al. [16] created three-dimensional hierarchical interlocked PVDF/ZnO nanofiber piezoelectric sensors by developing ZnO nanorods epitaxially on PVDF nanofibers using an electrostatic spinning process, resulting in excellent flexibility and high permeability. The combined piezoelectric effect of perfectly deformed ZnO nanorods that fit together and uniformly aligned PVDF nanofibers with highly electrically active phases made the material six times more sensitive in compression mode and forty-one times more sensitive in bending mode compared to pure PVDF nanofibers. Deng et al. [93] made a bendable piezoelectric sensor with cowpea-shaped PVDF/ZnO nanofibers and an electrostatic spinning method (CPZNs). This sensor demonstrated remarkable bending sensitivity due to the synergistic piezoelectric effect of the hybrid PVDF/ZnO and the polymer’s flexibility. It worked well in both compression and bending modes, with response times of 16 ms for compression and 76 ms for bending. The best compression and bending sensitivities were 0.33 V kPa^−1^ and 4.4 mV deg^−1^, respectively.

#### 2.3.2. Piezoelectric Hydrogels

Hydrogel is a type of polymer that has hydrophilic groups and a cross-linked three-dimensional network structure. This makes it impermeable to water and able to swell up in wet conditions. It has the capacity to absorb substantial amounts of water, leading to its inflation, and it boasts a high water content, all while retaining a soft texture and a well-defined form [97,98]. Hydrogel, characterized by its high water content and mechanical properties akin to skin tissue, is an optimal material for wearable devices. Researchers are trying to mix piezoelectric materials with hydrogel to make a composite hydrogel that has piezoelectric effects and can be used for tactile sensors that sense pressure [99,100,101].

In order to make composite piezoelectric hydrogels, piezoelectric material particles or fibers are usually mixed into a hydrogel matrix. This can also be achieved by mixing the piezoelectric material with the hydrogel solution and then letting the mixture cure or gel. To obtain the best piezoelectric properties, it is important to make sure that the piezoelectric material is evenly spread out in the hydrogel matrix during the preparation process. Fu et al. [94] developed a tough-powered hydrogel based on a polyacrylonitrile (PAN) hydrogel incorporating ferroelectric poly(vinylidene fluoride) (PAN-PVDF), where dipole interactions between the PVDF and the CN groups of the PAN chains promoted the highly electrically active β-phase from 0 to 91.3% crystallization, which yields the piezoelectricity of the PAN-PVDF hydrogel without the need for any electrodeposition process. They also obtained excellent properties such as skin-like Young’s modulus (1.33–4.24 MPa), tensile (175%), and high toughness (1.23 MJ/m^2^). Hu et al. [95] discussed a composite hydrogel that can be used to make flexible strain sensors that can sense both piezoresistivity and piezoelectricity. Cross-linked chitosan quaternary ammonium salt (CHACC) was used as the gel matrix, PEDOT:PSS was used as the conductive filler, and PVDF-TrFE was used as the piezoelectric filler. CHACC/PEDOT:PSS/PVDF-TrFE hydrogels were synthesized using one-pot and solution exchange methods. The hydrogel-based strain sensor exhibited very high sensitivity (GF: 19.3), fast response (response time: 63.2 ms), and a wide frequency range (response frequency: 5–25 Hz), while maintaining excellent mechanical properties (elongation at breakup to 293%). The incorporation of PVDF-TrFE attributed the enhanced strain sensing properties of the hydrogel to a greater change in relative resistance under stress and a wider response to dynamic and static stimuli.

Additionally, researchers have discovered that many natural materials are both environmentally friendly and biocompatible, but their piezoelectric properties are weak. Therefore, combining these materials with piezoelectric materials to create composite piezoelectric hydrogels has become ideal [102,103]. Gogurla et al. [72] made a soft, biocompatible, and skin-adhesive filamentous hydrogel by adding ZnO nanorods (ZnONRs), as shown in Figure 2f. When ZnONRs were added, the piezoelectricity was eight times higher than in the original silk hydrogel. When encapsulated in the silk protein layer, the hybrid effect of friction and piezoelectricity can potentially triple the electrical response. The high power generated (~1 mWcm^−2^) is sufficient to activate low-power electrical devices such as LEDs, oximeters, and stopwatches. Moreover, energy-generating skin (EG skin) can function as a haptic recognizer for finger movements, providing quantifiable real-time electrical signals. Composite piezoelectric hydrogels possess distinctive features and extensive uses; ongoing research will further enhance their potential across numerous domains. In the future, researchers will persist in investigating more application options and refining the preparation process and features.

## 3. Structure of Tactile Sensors

The structural design of tactile sensors is critical to their performance, particularly their sensitivity and response speed. In the development process, the introduction of bionic structures has brought innovative ideas to their design, taking advantage of biological sensing systems in nature and aiming to mimic the sensing structure of human skin or other animals to improve the adaptability and performance of tactile sensors [104,105], but this has made it difficult to classify the structure of sensors. As a result, tactile sensors are often divided into two categories based on the size of their structure: millimeter-scale structures and micrometer-scale structures.

### 3.1. Millimetre Structure

Both island–bridge and fork–finger structures common piezoelectric tactile sensors designed on a millimeter scale. Rigid islands dispersed on a flexible substrate (the sensor units) form the island–bridge structure, with a bridge connecting these islands. This design enables the sensor to deform under pressure and detect haptic information, as the combination of rigid and flexible structures enables it to adapt to various surfaces. The fork–finger structure is made up of alternating conductive fork–finger electrodes that introduce a periodic voltage across the piezoelectric material and make an electric charge when the material is pressed mechanically. The goal of these structures is to improve the sensor’s sensitivity and spatial resolution so that it can precisely identify and react to changes in external pressure.

The fork–finger electrode design offers an expanded surface area for enhanced contact with the measured object, hence augmenting the sensor’s sensitivity and measurement precision. Simultaneously, it can swiftly detect signal changes and rapidly transform them into electrical signals for processing, enabling the sensor to identify and measure the target item more promptly. Due to their simple design, high sensitivity, and strong stability, fork–finger electrode sensors serve as the foundational element of tactile sensors. Yan et al. [106] created a new type of piezoelectric bending sensor that has a 3D-crossed fork–finger microelectrode (Ag nanowires) inside a piezoelectric polymer film (PVDF-TrFE), as shown in Figure 3a. The piezoelectric film featuring embedded 3D-structured fork–finger electrodes attains a significant anisotropy coefficient due to the two piezoelectric modes (*d*_31_ and *d*_33_), which enhance its sensitivity to various bending orientations. The newly designed all-in-one piezoelectric multidirectional bend sensor has markedly enhanced mechanical stability, surpassing that of current piezoelectric bend sensors. The capacity to remotely manipulate the multidirectional movement of a smart car by only affixing an all-in-one flexible bending sensor to the wrist indicates that the sensor possesses significant promise for applications related to human–computer interaction.

On the other hand, the island–bridge structure typically consists of rigid bodies dispersed on a flexible substrate. Piezoelectric sensing units, located between the rigid array units and the flexible lower substrate, sense the magnitude of the pressure through the transfer of contact stresses. The island–bridge structure offers advantages in sensor design, such as high sensitivity, good adaptability, light weight, replaceability, and cushioning for energy absorption, which makes it promising for a wide range of applications in the field of flexible sensors. Lin et al. [107] suggested a new island–bridge piezoelectric flexible multifunctional haptic sensor array, as shown in Figure 3b. It is made up of three layers, among which are two PDMS membranes for protection, two PVDF membranes for sensing, and one PDMS membrane for insulation. A row + column electrode structure can be used with a small amount of wiring to add more pixels (m + n + 2 for n * m sensing pixels) and can sense and tell the difference between different external stimuli in real time based on their size, position, and pattern. Zhen et al. [108] developed a 3 * 3 flexible piezoelectric sensor array with a hybrid rigid–flex structure, as shown in Figure 3c. Devices with rigid–soft hybrid structures integrate small-pixel, high-sensitivity piezoelectric ceramic thick films, achieving satisfactory spatial resolution (6 mm), high sensitivity (15.08 mV/kPa), and optimal flexibility (4.23 mm bending radius). They also show strong applicability in joint motion monitoring, standing posture, and speech recognition. Speech recognition using an integrated convolutional neural network has an accuracy of 98.18%.

**Figure 3 sensors-25-00964-f003:**
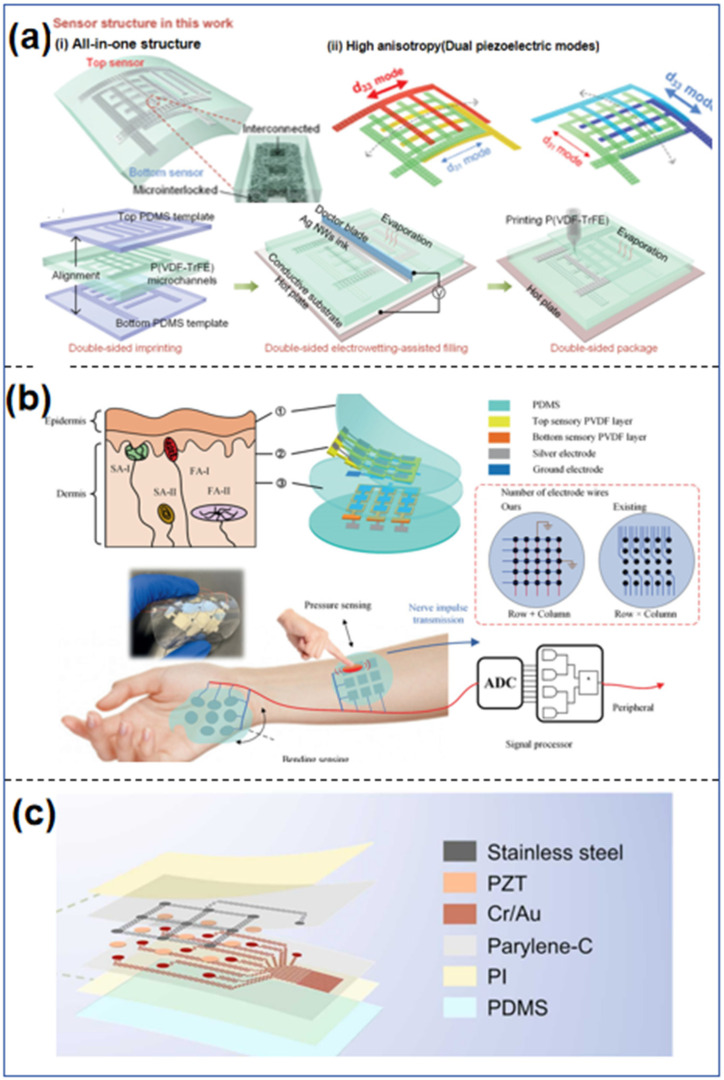
(**a**) Schematic of the structural design of the all-in-one piezoelectric multidirectional bending sensor [106]. (**b**) Schematic of the skin-inspired piezoelectric tactile sensor array [107]. (**c**) Schematic of the fabrication process of the sensor array [108].

### 3.2. Micron/Nano-Scale Structures

Precision machining techniques frequently fabricate micron-scale piezoelectric tactile sensors to achieve improved sensitivity and spatial resolution. These sensors are made up of precise microstructures made of micron-sized piezoelectric materials like PVDF or PZT that are glued to rigid or flexible substrates and wired to electrodes. This creates piezoelectric elements that turn mechanical pressure into electrical signals. A protective layer encases the entire structure, ensuring stability and durability, while enabling the sensor to interact with the external world for precise haptic feedback.

The micro-dome structure improves the sensor’s response to pressure fluctuations because of its unique geometry; hence, it increases sensitivity. It substantially improves the crosstalk resistance of the sensor array [32,109]. Zhang et al. [110] created a three-layer rigid–flexible hybrid haptic sensor array with a dome-shaped rigid–flexible hybrid force-transporting layer and a soft substrate, as shown in Figure 4a. This improves force transmission and causes noticeable amplification effects in the piezoelectric sensing layer’s *d*_31_ operational mode, as opposed to the usual *d*_33_ mode. The sensor demonstrates an ultra-high sensitivity of 346.5 pCN^−1^ (@30 Hz), a wide bandwidth of 5–600 Hz, and a linear force detection range of 0.009–4.3 N, surpassing the theoretical sensitivity of the *d*_33_ mode by 17 times. Furthermore, the sensor can detect multiple force directions by analyzing the outputs of four piezoelectric capacitors with high reliability, showing great potential for dynamic haptic sensing in robotics. The micro-dome structure makes sensors much more sensitive because of its unique geometrical benefits, larger contact area, high array uniformity, and ability to heal itself. This makes it a good choice for many uses in flexible electronics and intelligent robotics.

The micro-nano pyramidal structure of the pressure-tactile sensors makes them more sensitive because the stress is not spread out evenly at the base of the pyramid and is concentrated at the top. This causes the structure to deform significantly and improves its sensing performance. The pyramid structure provides advantages such as excellent homogeneity, stability, and stress concentration effects, which have led to its extensive use in sensor design, particularly in tactile sensors [111,112]. Seong et al. [113] successfully created a high-performance sensor with a pyramidal shape made of PDMS frictional material and barium titanate (BaTiO_3_) piezoelectric material, as shown in Figure 4b. The integration of BaTiO_3_ nanoparticles into pyramidal polydimethylsiloxane (PDMS) elastomers exhibited a synergistic combination of piezoelectric and triboelectric effects. The experimental results suggest that integrating piezoelectric materials with micropatterning techniques can significantly enhance the sensor’s performance. Hu et al. [114] developed a highly sensitive pressure sensor that consists of a polyimide substrate with high-density micropyramid arrays (HD-μPAs), a poly(vinylidenefluoride-co-trifluoroethylene) (PVDF-TrFE)/BaTiO_3_ (BTO) composite active piezoelectric element, and silver nanowires (AgNWs) as the top electrode, as shown in Figure 4c. The stress concentration effect, resulting from the large modulus mismatch between the nanofiber mats and the HD-μPA-integrated PI substrate, enhanced the piezoelectric performance.

The pyramidal structure, as a microstructure with high symmetry and complex geometry, is capable of producing a significant deformation response to external forces and exhibits good flexoelectric effects. The essence of the bending electrical effect is the generation of an electric dipole moment or field due to the deformation gradient when the material is bent. This effect is not only present in conventional piezoelectric materials but can also occur in some non-piezoelectric materials. It is closely related to the strain distribution of the material, the crystal symmetry of the material, etc. The geometric properties of pyramidal structures can significantly enhance the flexoelectric response of a material during deformation, and such structures can help improve the performance of sensors, the sensitivity of micromechanical systems, and other technological applications based on the flexoelectric effect. See Table 2.

**Figure 4 sensors-25-00964-f004:**
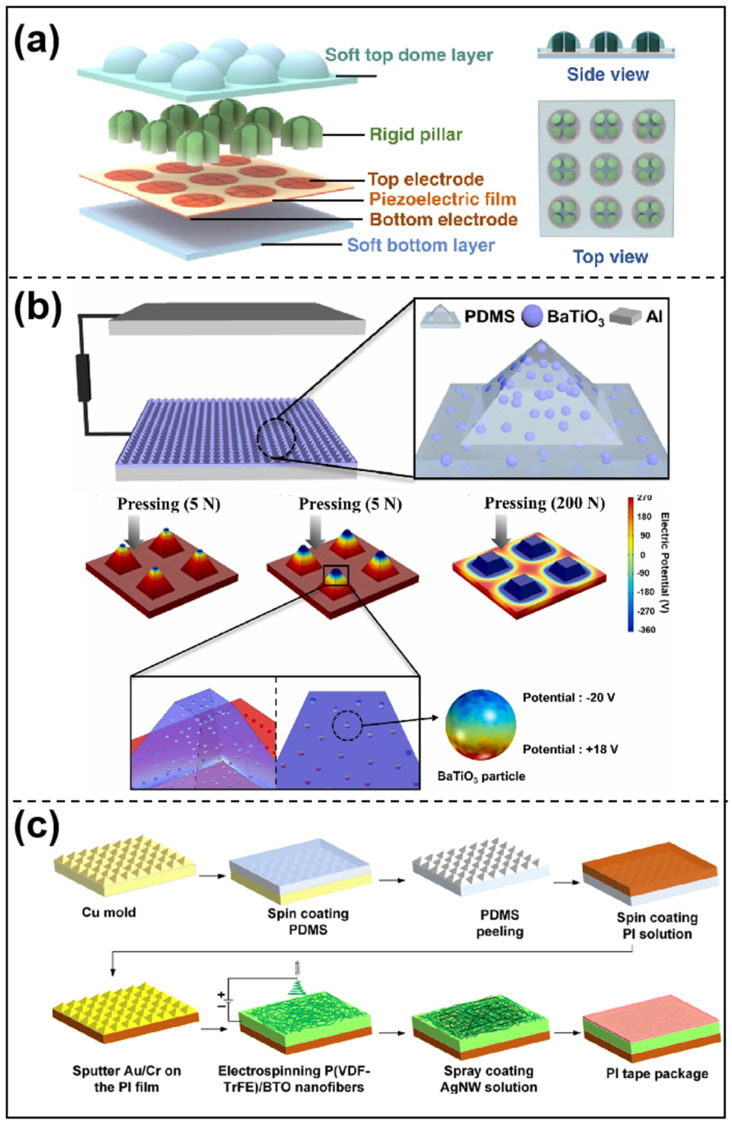
(**a**) Finger-excited soft–hard hybrid piezotactile sensor array [110]. (**b**) Schematic of BaTiO_3_ nanoparticles embedded in a pyramid−structured sensor [113]. (**c**) Fabrication process of HD−μPA integrated substrate pressure sensor [114].

**Table 2 sensors-25-00964-t002:** Features of different structures.

Structure		Specificities	Advantages	Disadvantages
Millimeter	Fork-finger structure	Usually has multiple branch-like structures.	Increase contact area and sensitivity.	The complexity of the structure leads to high costs.
			Suitable for application scenarios with large contact areas.	Sensitive to changes in loading points.
	Island–bridge structure	Connecting rigid electronic components (islands) to flexible parts (bridges).	Significantly improves the stretchability of the sensor.	The design and manufacturing process is more complex.
			Strong stability and increased life expectancy.	More sensitive to temperature changes.
Micron/ Nano-scale	Micro-dome structure	Improvement of sensor sensitivity and performance through the introduction of micro-nanostructures.	Fast response/recovery characteristics	Micro-nano structures are costly and technically complex.
			Wide bandwidth and linear force detection range	Relatively homogeneous structure, which does not allow for a hierarchical structural design.
	Pyramid structure	Capable of detecting a wide range of touches (pressure, shear, and torsion).	Increase contact area and sensitivity.	More complex design required for large range measurements
			Suitable for application scenarios with large contact areas.	Sensitive to temperature changes, signal stability and accuracy need to be improved

## 4. Fabrication of Tactile Sensors

The human skin is a multifaceted organ consisting of the epidermis, dermis, and subcutaneous tissue. The epidermis primarily serves a protective function, whereas the dermis is abundant in sensory nerve endings, which are essential for the skin’s ability to detect minor alterations. To attain high precision resolution in haptic sensors, both meticulous structural design and advanced processing techniques are essential. The primary techniques for fabricating piezoelectric tactile sensors encompass electrostatic spinning, 3D printing, and screen printing, among others.

### 4.1. Electrostatic Spinning

The fabrication of flexible tactile sensors employs electrostatic spinning technology. Many domains, including functional textiles and electronic devices, find extensive applications for electrostatic spinning, it being an effective, straightforward, and adaptable method for producing nanofibrous materials [115,116]. The fundamental principle involves elongating a charged polymer solution or melting it into fibrous material under an electrostatic field, thereby producing nanofibrous materials with distinct shapes and properties by adjusting various parameters. Electrostatic spinning makes nanofibers with a lot of specific surface area and holes in them. This makes it easier for the sensing material to stick to the target object, which improves the performance of the sensor [117,118].

The electrostatic spinning method is frequently employed to produce PVDF nanofibers, which possess great sensitivity, exceptional stretchability, and chemical stability, thereby fulfilling the requirements of flexible touch sensors. Lu et al. [119] introduced piezoelectric microfibers with a novel core–sheath structure, which were prepared by spinning polyvinylidene fluoride-trifluoroethylene (PVDF-TrFE) directly onto flexible wires using an electrostatic spinning method. As it is a simple method, we can achieve precise control over the fiber diameter and thickness of the functional layer of PVDF-TrFE. With a high sensitivity of 60.82 mVN^−1^ under positive pressure and a durability of 15,000 cycles, the sewable piezoelectric fibers can withstand a variety of complex surfaces and even severe deformations like bending and kinking, enabling their integration into textiles that promote breathability and comfort. Singh et al. [120] looked into how eight important factors affected the β-phase content in polyvinylidene fluoride nanofibers while they were being electrostatically spun, as shown in Figure 5a. The parameters were classified into four groups according to their role in electrospinning, and the grouped parameters were studied. These factors—rotational speed, electric field, and spin distance of the collector—have the most significant impact on the percentage of β-phase. They are the ones that mostly change how the polyvinylidene fluoride nanofibers stretch. This study provides an understanding of the influence of all important parameters in the electrospinning process. The article also describes in detail the development of PVDF nanofiber sensors based on electrostatic spinning. Overall, the electrostatic spinning technique plays an important role in the preparation of haptic sensors and enables the preparation of high-performance nanofiber materials, which improves the sensitivity and performance of the sensors.

**Figure 5 sensors-25-00964-f005:**
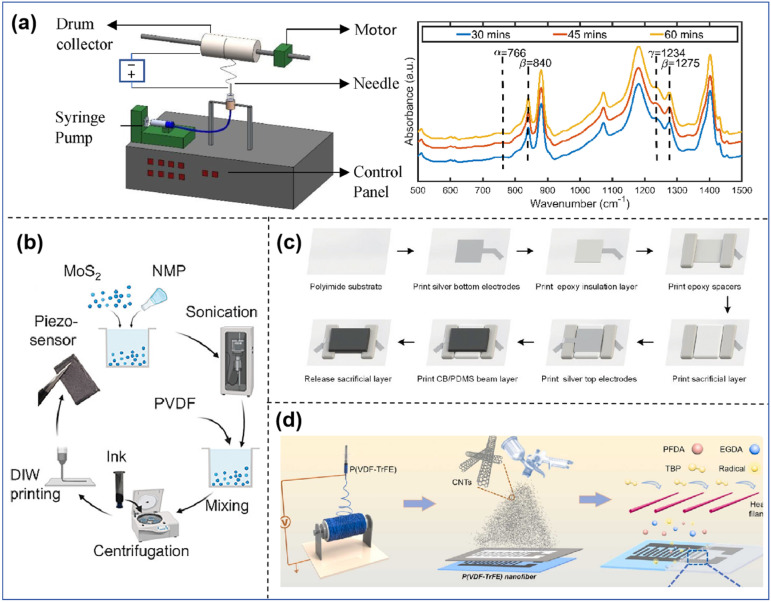
(**a**) Schematic diagram of the electrostatic spinning device used and FTIR plots of PVDF polyvinylidene fluoride samples prepared with different electrospinning times [120]. (**b**) Ink formulation and 3D printing process for PVDF and PVDF−MoS_2_ nanocomposites [121]. (**c**) Flowchart of TSA fabrication using a screen printing technique combining multilayer printing and the sacrificial layer technique [122]. (**d**) Electrode fabrication process [123].

### 4.2. 3D Printing

Three-dimensional printing technology plays a significant role in the preparation of tactile sensors, offering a flexible and efficient manufacturing method that diversifies and personalizes the preparation process. The combination of 3D printing and flexible sensing technology can foster the future development of biomedicine, artificial intelligence, and other fields [124,125]. First, computer-aided design software designs the sensor’s structure and then selects suitable printing materials. A 3D printer fabricates the sensor structure according to the design file, perhaps utilizing methods such as photopolymerization and material extrusion. Post-processing tasks, like the removal of support structures and heat treatment, may be necessary after printing to enhance sensor performance [126]. We ultimately evaluate the printed sensor to confirm its performance aligns with expectations and modify the design or printing parameters as needed based on the test outcomes.

The parameter settings of 3D printing have a direct impact on the sensitivity and performance of the haptic sensor, as the haptic sensor relies on the conductivity of the material, the fineness of the structure, and the responsiveness of the sensor as a whole, and the settings of specific parameters play an important role in the performance, including the printing material [127], the printing accuracy (layer height and resolution) [128], the printing speed, and the temperature control. Chang et al. [129] presented the material and structural design of flexible piezoelectric composites that exhibit a high electromechanical piezoelectric response. They used optimal 3D printing parameters with a DLP-based 3D printer, based on the formulation and structural design of the composites. The experiments showed that using a silane coupling agent and a dispersant was the best way to make 3D-printable, functionalized micro PMN-PZT ceramic–polymer composites. These composites had strong interfacial bonding, excellent dispersive stability, low viscosity, a smooth surface, high piezoelectricity, and excellent processing properties. The work of Islam et al. [121] greatly improved the piezoelectric properties of poly(vinylidene fluoride) (PVDF) by aligning dipoles inside PVDF-2D molybdenum disulphide (2D MoS_2_) composites while they were being 3D printed, as shown in Figure 5b. Shear stress in PVDF was employed to produce dipole polarization and align 2D MoS_2_ during 3D printing, enhancing piezoelectricity without requiring a post-polarization technique. The findings indicate a substantial enhancement in the piezoelectric coefficient (d_33_) above a factor of 8. The results indicate that 3D printing technology presents a highly promising method for fabricating high-performance piezoelectric devices and hence advancing the development of piezoelectric tactile sensors.

### 4.3. Screen Printing

Screen printing is pivotal in the fabrication of flexible tactile sensors and offers numerous advantages. It facilitates extensive batch manufacturing while ensuring elevated efficiency and minimal expense. Screen printing’s capacity to apply larger material layers and the utilization of screen masks enhance multilayer alignment, benefiting the fabrication of sensors with multilayer configurations [130]. The benefits of screen printing for tactile sensor fabrication encompass fewer limitations regarding substrate materials and printing pastes, rendering it appropriate for both rigid substrates and flexible films [131]. Different materials can serve as the functional fillers in the printing slurry. This simplifies the planar printing process, enabling the creation of devices that are multifunctional, heterogeneous, and heterostructured.

Researchers developed advanced piezoelectric tactile sensors with screen-printing technology. Wang et al. [122] made a fully printed tactile sensor array based on fingerprints, as shown in Figure 5c. They employed advanced screen printing techniques to identify and reconstruct microscale structures in three dimensions, utilizing various planar printing methods, multilayer printing techniques, and sacrificial layer techniques. This configuration enabled the tactile sensor array to gather comprehensive geometric data about the microtexture, leading to the creation of a Braille recognition system using this tactile sensor array. Jo et al. [132] employed screen printing to create sensors from PZT-PVB composites of grades 0–3 on two distinct substrates: alumina film and polyester film. They then compared the sensors printed on different substrates and at different temperatures. The piezoelectric coefficients (*d*_33_, *g*_33_) of sensors polarized at various temperatures were compared, leading to the experimental conclusion that the 0–3 PZT-PVB composites produced via screen printing exhibit high sensitivity and rapid response time, rendering them suitable for tactile sensors.

### 4.4. Spray/Spin Coating

Spray coating technology, which atomizes and deposits conductive, piezoelectric, or other functional materials onto a flexible substrate to create the sensor’s functional layer, is a prevalent technique for fabricating flexible tactile sensors. This technique facilitates extensive, uniform coatings and is appropriate for fabricating sensors with intricate geometries [133,134].

The spraying process enables precise control over the thickness and content of the coating, thereby influencing the sensor’s performance. Su et al. [123] developed a superhydrophobic antimicrobial self-powered PENG sensor, as shown in Figure 5d. They used an electrostatically spun PVT nanofiber membrane, which is a pressure-sensitive material, to form the sensor with sprayed forked carbon nanotube bipolar electrodes and then encapsulated it with PFDA-coEGDA nanocoatings via initiated chemical vapor deposition (iCVD). The sensor can work in a wide range of harsh environments thanks to its conformal hydrophobic iCVD nanocoating, which makes it mechanically strong and chemically stable. The robustness of the sensor enables an extremely wide range of applications, such as integrating it into a shoe insole to monitor the human gait in real time. Ding et al. [135] fabricated a novel ultrasensitive hydrogel tactile sensor based on asymmetric ionic charge injection using spray technology. These devices have an external operating voltage of only a few tens of millivolts, an extremely low detection force of 0.075 Pa, a sensitivity of 57–171 kPa^−1^, and excellent cyclic reliability during pressing. The sensors offer ultra-high sensitivity, ultra-low detection limits, low operating voltage, and long life, enabling accurate recording of pulse and acoustic pressure signals. See Table 3. This article uses the number of ★ to indicate how much the cost is, the more ★ the higher the cost, the less ★ the lower the cost.

**Table 3 sensors-25-00964-t003:** Comparison of various processes.

Technology	Advantages	Disadvantages	Costs
Electrostatic spinning	Preparation of materials with high surface area and porous structure.	Low mechanical flexibility.	★★★
	Enhanced physical performance of the sensor.	High cost and poor stability.	
3D printing	Rapid production of complex shapes.	Limited choice of materials.	★★★★
	Material and process flexibility.	The connection is prone to problems.	
Screen printing	Large batch size, high efficiency, and low cost.	Thick-film high-precision printing still needs further improvement.	★
	Multilayer alignment is easier to achieve.	Ink may pass through the screen and cause unclear patterns.	
Spray/spin coating	Fast realization of large coating areas.	There are certain requirements for the viscosity and solids content of the solution.	★★
	Easy operation.	May produce uneven thickness	

## 5. Principal Applications

In the current age of swift technological advancement, the utilization of flexible tactile sensors is increasingly infiltrating various facets of our existence, encompassing medical and health monitoring, intelligent robotics, and wearable technologies. These advanced sensors, characterized by their exceptional flexibility, sensitivity, and adaptability, have initiated a new age in human–machine interaction [136,137,138]. They can replicate the tactile functions of human skin, detecting external contact, pressure, and temperature variations while also delivering accurate feedback in diverse complex situations. The subsequent parts of this paper will examine the varied applications of flexible tactile sensors, demonstrate their critical importance in several domains of contemporary technology, and analyze future prospects in this field.

### 5.1. Human Health Monitoring

Flexible haptic sensors are extensively utilized in medical monitoring applications for tracking physiological signals, including heartbeat, respiration, blood pressure, and pulse, owing to their capacity to detect subtle stressors with great sensitivity while maintaining flexibility, non-invasiveness, and comfort [139]. Flexible patches affixed to human skin exhibit significant potential for medical monitoring, particularly in the assessment of pulse waveforms and blood pressure [140]. Zou et al. [141] showed a self-powered, flexible arch device for pulse sensing, as shown in Figure 6a. They used a PVDF film inside a self-arch structure to make a hybrid nanogenerator that uses both triboelectric and piezoelectric effects to improve the output, signal-to-noise ratio, and stability during pulse signal detection. Affixed to the wrist, the gadget monitors the pulse signal, transforming it into an electrical signal that reflects the physiological attributes of the human body. Numerous studies have established a correlation between pulse waves and blood pressure via algorithms that facilitate blood pressure measurement.

Health monitoring devices need to consider longevity, Yi et al. [142] created a wireless wearable system for continuously monitoring blood pressure, as shown in Figure 6b. It has two piezoelectric sensors, PZT piezoelectric thin film membranes, and a wireless data acquisition module to measure the arterial pulse at the lateral epicondyle and the belly of the index finger. Using an arterial pulse piezoelectric response reduces motion artifacts caused by the raw arterial pulse wave being specific to posture. This makes this method suitable for continuous blood pressure monitoring in wearable tech. To solve the problem of measurement accuracy during motion, inspired by the mechanoreceptors (MD and cilia) of human skin, Chun et al. [86] developed a self-powered, stretchable, and wearable gel mechanosensor, as shown in Figure 6c. The fabricated sensors can be fully attached to the skin and measure a variety of kinematic signals (e.g., muscle movement, radial artery BP waveforms, and body movement), and, in particular, BP pulse behavior was shown to be measurable even after moving the muscle.

**Figure 6 sensors-25-00964-f006:**
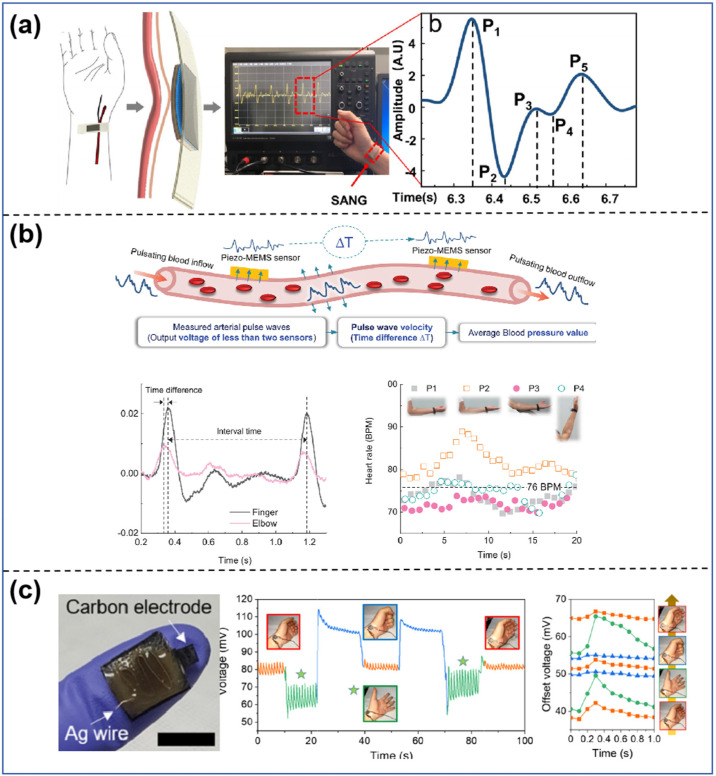
(**a**) Schematic and arch construction picture of the sensor and the detected radial artery pulse waveform [130]. (**b**) Principle of sensor operation and interval between two neighboring pulses and expected heart rate [131]. (**c**) Schematic of the sensor structure and the BP waveform generated by contracting the wrist muscles when placed in the wrist position [77].

### 5.2. Motion Monitoring

Flexible tactile sensors exhibit remarkable flexibility and ductility, enabling them to adhere closely to the skin’s surface for the monitoring and recording of human movement data. They achieve this by sensing and measuring subtle changes in the skin and muscles during movement while simultaneously detecting various movement patterns in real time, including walking, jumping, and running [15,143,144]. They can also assess the athlete’s posture and adjust their workout plan in real time through feedback, thereby providing accurate biomechanical information for various fields such as movement analysis, health monitoring, rehabilitation training, and others [145]. Zhu et al. [146] created an innovative, flexible, elastic, self-healing composite nanogenerator for monitoring human motion sensors, as shown in Figure 7a. The assembly comprises PENG, TENG, and hydrogel electrodes, utilizing PVDF as the pressure-sensitive material. The integration of the piezoelectric effect and the triboelectric effect enhances both sensitivity and measuring range. The good tensile and self-powered properties of the hydrogel electrode meet the demands of strenuous exercise, enabling it to monitor multidimensional human body movements like bending, twisting, and rotating, including the spiral pulling action of table tennis and the 301 C technique of diving.

The incorporation of touch sensors into sporting equipment enhances the monitoring of motor information; hence, it refines motor skills. Yao et al. [147] suggested designing and 3D printing a group of flexible piezoelectric nanocomposites that have good functional responsiveness and a lot of different structural features that can be changed, as shown in Figure 7b. They 3D-printed a flexible piezoelectric lattice, characterized by a stretch-dominated micro-architecture and a specified thickness of 5 mm, using trial-koxysilane-methacrylate surface modification of PZT; they then incorporated it into boxing gloves. Using supplementary data, coaches can instruct boxers on their motions with the help of a customized user interface that displays the force distribution, data-collecting, and wireless transmission modules. Chen et al. [148] developed a machine learning-augmented intelligent tennis training system using self-powered sensing yarn arrays, as shown in Figure 7c. They employed a simple and effective method to combine piezoelectric nanofibers and triboelectric materials into a single yarn, enabling the simultaneous conversion of frictional and piezoelectric signals. The training system comprises a smart elbow pad, a smart tennis racket, a machine learning algorithm, and a feedback application. The system provides athletes with accurate guidance and support, offers fresh perspectives, and expands the boundaries of intelligent sports, thereby expanding the field of smart sports training.

**Figure 7 sensors-25-00964-f007:**
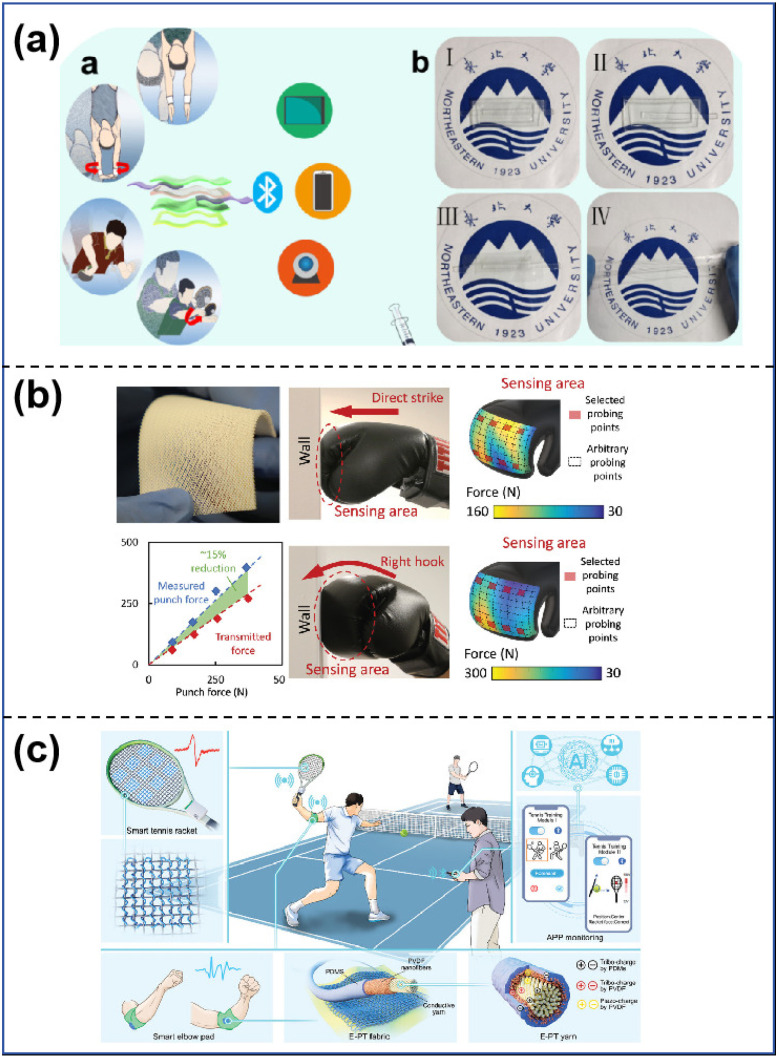
(**a**) Mapping of the sensor and its application [146]. (**b**) Optical image of a flexible self-sensing lattice and its application [147]. (**c**) Schematic diagram of the smart tennis training system [148].

### 5.3. Human–Computer Interaction

Human–computer interaction (HCI), facilitating human control over hardware and the acquisition of feedback, serves as a conduit between humans and machines, garnering significant attention and undergoing quick advancement in recent years [136,149]. Flexible tactile sensors are mostly used in human–computer interaction because they can mimic the way human skin senses touch, which lets robots and other smart devices feel like they are touching a person. This facilitates precise identification and response to physical attributes such as shape, texture, temperature, and pressure, thereby augmenting the naturalness and intuitiveness of interactions and enhancing the user experience [150,151,152,153].

The perception of objects’ form and roughness, soft robotics, bending degree, and gesture identification are critical components of human–computer interaction. Gao et al. [152] introduced a system that integrates a flexible PVDF-TrFE piezoelectric haptic sensor with a high-conductivity hydrogel piezoresistive sensor. The piezoresistive sensor effectively detects static strains to delineate an object’s contour, while the piezoelectric sensor responds to dynamic strains to capture surface information. When integrated with machine learning, this facilitates a more precise identification of the object’s contour. Zhu et al. [154] introduced a smart glove featuring a piezoelectric friction electroactive sensor and a PZT piezoelectric tactile mechanical stimulator, as shown in Figure 8a. The suggested smart glove enables sophisticated joint manipulation and relays real-time impact events via piezoelectric haptic stimulation, enhancing the experience of indirect engagement. This will also enhance the rehabilitation of impaired people. Shu et al. [104] made a shape-sensing electronic skin (SSES) using PVDF and a differential piezoelectric matrix, as shown in Figure 8b. This allows soft robots to have both proprioceptive and exteroceptive functions by using machine learning techniques. They can construct a dynamic model using the acquired form fluctuations, achieving a resolution of 0.0025° and a response time of 36 ms, thereby endowing the soft robot with exteroceptive abilities and fundamental intelligence. Yan et al. [155] introduced a novel piezoelectric film design with 3D-structured microelectrodes, as shown in Figure 8c. They achieved this by electro-wetting conductive nano-inks, which enabled printing onto pre-formed mesh microchannels in the piezoelectric film. Putting a high-performance piezoelectric device on a person’s finger to control a robot’s movements from a distance using a human–computer interface shows how useful our piezoelectric film could be in real life.

Piezoelectric tactile sensors have a wide range of applications not only in these fields, but also in the industrial sector, mainly for improving automation and accuracy, thanks to their excellent properties, including resistance to bending, withstanding a wide range of temperatures and humidity. In robotics, piezoelectric sensors help robots to sense tactile feedback and precisely control assembly and handling processes. In smart manufacturing, piezoelectric sensors monitor pressure and haptic changes on production lines in real time to optimize product quality control and machining accuracy. They can also be used for precision quality inspection of workpieces to improve inspection accuracy and detect potential problems in a timely manner. In human–machine interfaces, piezoelectric sensors provide tactile feedback to operators to enhance the operating experience and work efficiency. Piezoelectric sensors are also widely used for non-destructive testing of multilayer material sheets. By monitoring small pressure changes on the surface or inside the material, they can effectively detect defects in the material, such as cracks or interlayer separation, thereby improving material safety and reliability. In conclusion, piezoelectric tactile sensors play an important role in improving the accuracy, reliability, and automation of industrial equipment.

**Figure 8 sensors-25-00964-f008:**
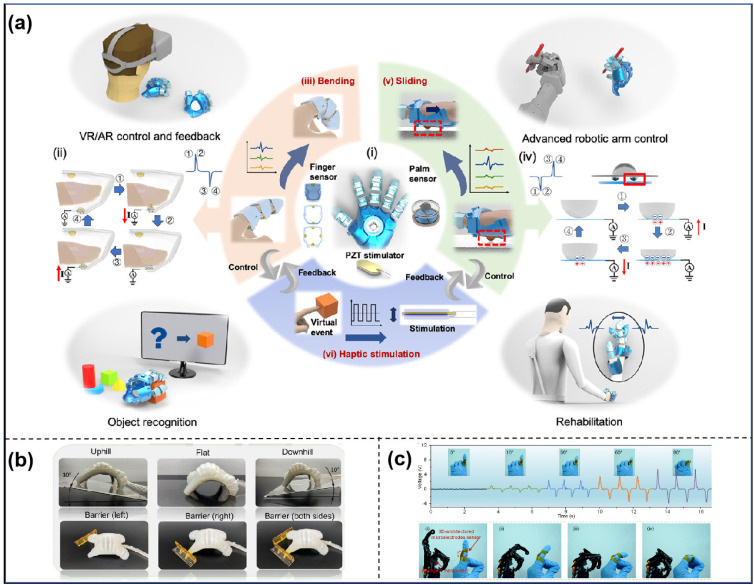
(**a**) Schematic diagram of a glove−based HMI for various applications [154]. (**b**) Recognition of five new terrains not yet trained using sensors [104]. (**c**) Example of sensor sensing bending angle and gesture remote control application [155].

## 6. Summary and Outlook

Flexible tactile sensors are characterized by flexibility, stability, ease of processing, and low cost; they have attracted much attention in the fields of wearable devices, medical monitoring, and artificial intelligence, etc. They not only enhance the naturalness and sensitivity of human–computer interaction but also promote the development of virtual reality, intelligent robotics and medical devices. The first part of this paper focuses on flexible piezoelectric tactile sensors and how piezoelectric materials are grouped. These materials include inorganic and organic piezoelectric materials as well as composites. We subsequently elucidate methodologies for preparing these materials and sensors, fostering a comprehensive understanding of them. Finally, we delineate the applications of flexible piezoelectric tactile sensors in human information monitoring and human–computer interaction.

Flexible piezoelectric tactile sensors have made breakthrough progress in research but still face many challenges in practical applications. Firstly, the performance of flexible piezoelectric materials is still inferior to that of traditional piezoelectric materials, and it is necessary to improve their performance through material design and modification, structural optimization, and dynamic regulation, all while maintaining flexibility. Second, the design of dense sensor arrays to enhance sensitivity and versatility is a growing trend, but at the same time there, are signal crosstalk problems; the main solutions to this problem are to optimize electrode design, use machine learning algorithms to decouple the signal, and use identification to improve the accuracy and efficiency of signal processing. Thirdly, the fit and stability of flexible sensors in situations with long-term wear and high-humidity environments is critical and can be achieved through the development of hydrogels, self-repairing materials, and combination with textile clothing. Finally, the problem of not achieving full flexibility can be addressed by investigating new flexible conductive materials such as silver nanowires, liquid metals, conductive polymers, and carbon-based materials for the design of flexible circuits. These challenges also present opportunities for development, driving material preparation, device processing, and system integration. In the future, piezoelectric flexible tactile sensors will be miniaturized, multifunctional, and integrated, and the boundaries of their use will be greatly expanded to play an irreplaceable role in more fields.

## Data Availability

Not applicable.

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
