# Peer review of "Recent Progress in Flexible Piezoelectric Tactile Sensors: Materials, Structures, Fabrication, and Application"

_sensors, 2025, doi:10.3390/s25030964_

Round 1
Reviewer 1 Report
Comments and Suggestions for Authors
Flexible tactile sensors have become a hot research topic as they are widely used in many fields. This review provides a comprehensive and in-depth overview of flexible piezoelectric tactile sensors. The authors integrate the latest research results and clearly describe the key technological challenges and future research directions. The authors demonstrate a deep understanding of the technical details of flexible tactile sensors, which is valuable for readers. I recommend this article for publication.
1. The first and second paragraphs of the introduction are a bit tedious, so please be brief.
2. The formatting in the images should be uniform and coordinated, e.g. (a), (b), etc. in Fig 5 and Fig7.
3. In the final outlook section, the problem should be described briefly and should be followed by possible directions for a solution.
Reviewer 2 Report
Comments and Suggestions for Authors
This study provide an ambitious goal of developing flexible tactile sensors. This review paper is generally well organized and effectively outlines the objective, approach, preparation methods, applications, and related studies on flexible tactile sensors. However, certain sections require further elaboration and modifications before I can recommend it for acceptance. My comments are outlined below:
1. In line 48, can you define what prizoresitive sensors are? How they can be produces? Their applications, and what is the different between piezoresitive and piezoelectric sensors? About the manufacturing and application of piezoresitive sensors you can indicated the following studies
-Impedance analysis for condition monitoring of single lap CNT-epoxy adhesive joint. International Journal of Adhesion and Adhesives, 2019, 88, 59-65.
-Recent progress in flexible and stretchable piezoresistive sensors and their applications. Journal of the Electrochemical Society, 167(3), 2020, 037561
-Structural health monitoring of defective single lap adhesive joints using graphene nanoplatelets. Journal of Manufacturing Processes, 2020, 55, 119-130.
2. In line 62, the authors mentioned the application of piezoelectric sensors in various industries such as medical, robotics, and e-skin. Please expand the discussion to include additional significant industries, such as composite structural health monitoring in aerospace and marine applications, and provide relevant references to support these applications.
3. In line 93, please provide examples of inorganic piezoelectric materials or compounds.
4. In line 102, "have made" instead of "has made". In line 106, Could you elaborate on the “few other piezoelectric crystals”? What are their distinguishing features, and why are they less commonly used than ZnO, AlN, and MoS2?
5. In line 137, can you characterise which oxides are usually used in piezoelectric ceramics and the typical temperature range for sintering?
6. In section 2.1.2, could you include a table comparing piezoelectric ceramics such as BaTiO3, PbTiO3, PZT, and Sm:PMN-PT? The comparison could highlight their sensitivity, advantages, applications, and limitations.
7. In section 4.2, please provide a more detailed explanation of the 3D printing manufacturing parameters involved in the fabrication of tactile sensors. Discuss how variations in these parameters can influence the sensitivity of tactile sensors. Please bring relevant references to support these descriptions.
8. In Section 5, Principle Applications, please provide a dedicated subsection on the use of flexible tactile sensors for structural health monitoring of composite structures or composite laminates in the various industries.
9. use bullet points to highlight your main results in the Conclusions section
Comments on the Quality of English LanguageThe English of the text should be double checked.
Reviewer 3 Report
Comments and Suggestions for Authors
The article is a review of 145 works devoted to one of the current and popular topics related to the development of the design, manufacture and application of flexible piezoelectric tactile sensors. The authors dwelled in detail on the review of piezoelectric materials for tactile sensors, their properties and advantages in solving certain problems. Particular attention is paid to the selection of the optimal design. In general, the article is of considerable interest to a wide range of readers, in particular specialists working in the field of control and monitoring of movement and the state of the human body. At the same time, it is necessary to note some unevenness in the presentation of the material. On the one hand, a detailed presentation of well-known information about piezoelectricity (section 2), on the other hand, the use of little-known special terms (for example, “electrostatic spinning” in sections 2.2.1, 2.3.1, etc.). Another drawback is the lack of information on the modeling of materials and the design of the sensors presented in the work. I believe that the article can be published after minor revision.
Reviewer 4 Report
Comments and Suggestions for Authors
1. I would recommend to point the necessary parameters of the tactile sensors in the table or write a few paragraphs with their applications.
2. Please provide the durability of such elements. Moreover, it would be helpful emphasize intrinsic conditions of the sensors (temperature, humidity)
3. Check the deciphering of some abbreviations in the text, e.g. iCVD, EG skin etc.
Reviewer 5 Report
Comments and Suggestions for Authors
This is a review paper focuses on the flexible piezoelectric tactile sensors. The paper was well written with a complete in structures. It can be accepted for publication after some minor revisions.
1. Some new comments regarding the importance of piezoelectric tactile sensor can be stated in the paper.
2. Section 2: It should be noticed that in some materials, including piezoelectric materials, the flexoelectricity may be occurs. Therefore, the authors are encouraged to discuss about this phenomenon.
3. Section 3: There is a new structure of tactile sensor named “Bio-inspired structures”. This structure can be used in advanced robotics, biomedical applications, and adaptive interfaces. It should be discussed in the manuscript.
4. Section 5: The applications of the tactile sensor in industry should be more discussed.
5. To prevent typos, the entire paper should be reread and corrected.
Round 2
Reviewer 2 Report
Comments and Suggestions for Authors
Accept in present form
Reviewer 5 Report
Comments and Suggestions for Authors
All issues were well addressed. It can be accepted in the present form.